# Meta-Analysis of Mitochondrial DNA Control Region Diversity to Shed Light on Phylogenetic Relationship and Demographic History of African Sheep (*Ovis aries*) Breeds

**DOI:** 10.3390/biology10080762

**Published:** 2021-08-10

**Authors:** George Wanjala, Zoltán Bagi, Szilvia Kusza

**Affiliations:** 1Centre for Agricultural Genomics and Biotechnology, H-4032 Egyetem tér 1, 4032 Debrecen, Hungary; geog.wanjala@gmail.com (G.W.); bagiz@agr.unideb.hu (Z.B.); 2Doctoral School of Animal Science, University of Debrecen, H-4032 Böszörményi út 138, 4032 Debrecen, Hungary

**Keywords:** genetic diversity, haplotypes, indigenous sheep, matrilineal lineage, mtDNA control region

## Abstract

**Simple Summary:**

The African continent is home to more than 400 million heads of sheep, the majority of which are classified as indigenous and raised primarily for subsistence. They live and thrive well in a wide range of climatic and production conditions, ranging from unfavorable to favorable environments. Recent molecular and archeological studies have hypothesized that these breeds harbor novel genomic regions that give them the ability to adapt to varied production environments. However, the genetic relationship among these populations is poorly understood. Knowledge about the population history and genetic relationships between populations provides an opportunity for the improvement of breeding and conservation programs. We meta-analyzed 399 African sheep breeds mtDNA control region sequences retrieved from the NCBI GenBank database to elucidate their diversity, phylogenetic relationship, and demographic history.

**Abstract:**

To improve sheep breeding and conservation of genetic resources, the mitochondrial DNA control region (mtDNA CR) of 399 sequences of African indigenous sheep breeds from previously published research articles were meta-analyzed to elucidate their phylogenetic relationship, diversity, and demographic history. A total of 272 haplotypes were found, of which 207 were unique and a high level of mtDNA CR variability was observed. Generally, the number of polymorphic sites, nucleotide and haplotype diversity were high (284, 0.254 ± 0.012 and 0.993 ± 0.002, respectively). The median-joining (MJ) network of haplotypes produced three major haplogroups (A, B and C), with haplogroup B being dominant. A mixture of populations suggests a common matrilineal origin and lack of and/or a weak phylogeographic structure. Mismatch analysis showed recent expansion of North African breeds, whereas East African and continental populations exhibited selection pressures for adaptation. A slight historical genetic difference was also observed between the fat tail and thin tail sheep breeds. However, further investigations are required using more samples and long sequence segments to achieve deeper levels of conclusions on the African sheep phylogenetic relationship. The present meta-analysis results contribute to the general understanding of African native sheep populations for improved management of sheep genetic resources.

## 1. Introduction

Domestic animal agriculture plays a crucial role in maintaining food security, as envisioned in sustainable development goal two (“End hunger, achieve food security and improved nutrition and promote sustainable agriculture”) [1]. At present, the estimated global sheep, goat and cattle populations are 1.2, 1.1, and 1.5 billion heads, respectively [2]. The African sheep population is estimated to be above 400 million heads. To enhance production, diversity and adaptation under the current climate change challenges [3] in recent decades, researchers have extensively studied the origin, history, phylogenetic relationships and diversity of domestic animals within species and among populations such as in sheep [4,5,6,7], goats [8,9,10], and cattle [11]. Information on the diversity and history of the species and/or breed provides the foundation on which conservation and breeding programs are made [12].

Archeological evidence suggests that sheep were among the first animals to be domesticated approximately more than 10,000 years ago in present Iran, some Arabic countries and Turkey, formerly referred to as the fertile crescent at the Zagros Mountains of Iran [9,13]. It is hypothesized that at least two independent domestication events took place, leading to the raise of haplogroup B and A which are globally most frequent [4,14]. According to studies, haplotypes belonging to haplogroup B are descendants of European mouflon (*Ovis aries musimon*) and are more frequent in Europe [4,6,14], while haplogroup A descends from Asiatic mouflon (*Ovis orientalis*) which is mainly present in Asian breeds [4,15]. Other authors, however, aver that European mouflon, a native to Sardinia and Corsica Islands in the Mediterranean Sea are pre-domesticated forms (feral) of Asian Mouflon [14,15]. Further studies have identified the existence of other sheep clades from C to E [4]. These could have risen from post domestication selection leading to the emergence of varied phenotypes, for instance, meat production traits, wool characteristics, coat color, presence or absence of horns, body sizes and tail size [15]. Studies indicate that haplogroup C is present in China [4], whereas clades D and E are very rare although spotted in near East sheep [14].

The history of domestic sheep in Africa is still inconclusive and thus it is an area calling for further investigation. This could be due to inadequate mtDNA CR studies done on African sheep breeds to explore their matrilineal lineages. However, the available inferences suggest that sheep spread throughout Africa with the pastoral communities after being introduced from the Near East [4]. The role of the East African region in the spread of sheep in the African continent is largely unknown, although, it is thought it was vital to the introduction of sheep from Southern Asia or the Arabian Peninsula and/diffusion southwards African continent [4,16].

Studies on the matrilineal lineage of African sheep breeds have yielded consistent results that clades A and B are the most frequent in the region, with clade B being the most common. Haplogroups C and E have been reported but not as common. For instance, it was reported that clade B is most common in some sheep breeds from Algeria [17], Ethiopia [18], South Africa [19], Egypt [5,20], Kenya [4] and Morocco [21]. These studies revealed that sheep breeds in Africa belong to different matrilineal lines, giving them an advantage of genetic variation between breeds and among populations, which is essential for improved production, environmental tolerance, and evolution. The few studies on the matrilineal lineage of African sheep breeds have documented how clade B arrived in Africa, however, the lack of documentation and evidence on other haplogroups is an area to be explored in future studies. It is worth noting that loss of genetic diversity within species and among populations is harmful not only in the context of conservation and current productivity but also for future utilization, as some genes lost at present might be vital in the future [18], under the dynamic global economic and social status and climate change as well. Perhaps it is also worth noting that species lose their genetic diversity through uncontrolled gene flow between and within populations leading to inbreeding, reduced effective population size, consequently causing inbreeding depression e.g., [22,23].

Population geneticists have utilized DNA polymorphisms to study the “true” genetic structure, relationships of livestock species, within and between populations, for example, by using microsatellite markers [24,25], SNPs [26,27]. mtDNA CR, also referred to as the displacement loop (D-loop) is the most popular DNA polymorphism used to study matrilineal lineages and population history. mtDNA CR is informative due to its low rate of recombination and maternal line inheritance [28], its rapid substitution rate compared to nuclear DNA [29], coupled with the ease and low costs of sequencing [30]. For instance, using mtDNA CR sequencing, the origin and history of several species have been studied [4,20,30,31,32] and the results have proved informative.

The African continent boasts of over 170 domestic sheep breeds, of which 80% are classified as native populations [33] produced under a traditional extensive system by resource-poor farm families and pastoralists. According to the authors, small ruminants in Africa are more valuable next to cattle, emphasizing their role in global food security i.e., access to nutritious food by all people [1]. Phenotypically, African sheep breeds are broadly classified into two groups; fat tail and thin tail. The origin of fat-tailed sheep is also not clear because the main sheep ancestor (*Ovis orientalis*) was thin-tailed. Interestingly, fat-tailed sheep are broadly distributed across the African continent and other parts of the world. This could be the evidence of post-domestication selection in response to human preference or harsh climatic conditions [16,34]. According to Moradi et al. [34], the fat-tail phenotype is deemed an adaptive response to unfavorable environmental conditions and a valuable energy reserve during migration and winter. Several other adaptive and productive phenotypes including inter alia coat type (wooled or hair), and short or tall legs were developed later [35].

Studies aiming to shed light on the genetic differentiation, phylogenetic relationships and matrilineal lineages in sheep breeds have been conducted worldwide, targeting specific breeds in specific countries. mtDNA CR markers have been instrumental in these studies. Similar but few studies have been conducted in Africa, with the main aims of assessing the evidence of domestication and inferring the ancestral origin of the sheep breeds by observing the haplogroups present in the populations. However, to the best of our knowledge, there is no literature exploring the phylogenetic relationships among sheep breeds in the African continent. As a result, the current meta-analysis aims to examine the mtDNA CR diversity and phylogenetic relationships between sheep populations in Africa by retrieving and analyzing the mtDNA control region sequences from the NCBI GenBank database [36]. Knowledge of the genetic relationship between sheep populations in African is vital in improving the conservation measures in the efforts of maintaining continental biodiversity as envisaged in sustainable development goals [37].

## 2. Methods

### Data Source

The 469 mtDNA CR sequences used in this meta-analysis were retrieved from the NCBI GenBank database [36] (see Appendix A). Research articles were used to identify the most relevant accession identification numbers (AID), specifically for sheep breeds from African Continent. The search for articles was performed on both google scholar and PubMed from 2005 to 2021 using the following terms “sheep” AND “goats mtDNA”, “genetic diversity“ OR “mtDNA diversity” OR “MtDNA” OR “Matrilineal lineage” OR “Origin” OR “domestication” followed by either Africa, African regions by name or specific African countries. The articles found were screened by title and abstract. Most relevant articles were scanned through manually for purposes of checking whether respective AIDs are indicated or not. Final papers that had AID for partial mtDNA D loop or control region sequences were selected, from which AID were extracted for further sequence retrieval from the NCBI GenBank database under nucleotide option considering all partial mtDNA sequences listed since 2005 to 2021. Figure 1 depicts the PRISMA flow chat [38] of the literature review process. 

The retrieved sequences were aligned and edited using the CLUSTALW [39] and MEGA X [40] software resulting in 399 homogenous sequences of 481 base pairs (from 15,640 to 16,121 bp) for downstream analyses.

DNA sequence polymorphism software (DnaSP v6) [41] was used to group the resultant sequences into regions representing North Africa, West Africa, East Africa, and South Africa populations. Furthermore, it was also used to calculate the overall sequence polymorphism including the number of polymorphic sites, number of haplotypes, haplotype polymorphism, nucleotide polymorphism, number of segregating sites following the procedures of Tajima [42] and Nei [43]. To determine the source of mtDNA and haplotype frequency differences, analysis of molecular variance (AMOVA) and pairwise Fst were calculated using Arlequin software v 3.5.2.2 [44]. Then, the program network 4.6.1.6 [45] was used to draw the median-joining (MJ) network from haplotypes generated in DnaSP v6. Further, the software Arlequin v 3.5.2.2 [44] was used to calculate neutrality indices (Tajima’s D and Fu’s Fs values) and to estimate mismatch distribution.

## 3. Results

### 3.1. mtDNA CR Polymorphism Indices

Representing four regions of the African continent, 399 mtDNA CR sequences of 481 bp length were analyzed. The summary of observed polymorphic sites is presented in Table 1. We observed a mean number of polymorphic loci of 284 (267 parsimony-informative and 17 singleton sites) which defined a total of 272 haplotypes. Overall, the average nucleotide (pi) and haplotype diversity (Hd) were high; 0.254 ± 0.012 and 0.993 ± 0.002 respectively. Region-wise, the haplotype diversity was high, ranging from 0.956 ± 0.059 for the South African to 0.998 ± 0.001 for North African sheep breeds. Similarly, nucleotide diversity (pi) and the average number of nucleotide differences (k) also varied significantly among populations. Specifically, pi ranged from 0.008 ± 0.001 to 0.263 ± 0.015 while k ranged from 3.615 to 116.994. The ratio of G+C nucleotide content also varied from 0.304 for North African to 0.418 for West African populations. Overall, the ratio of G+C recorded was 0.321.

### 3.2. Phylogenetic Relationship

Out of all 272 haplotypes observed, 76% were unique, suggesting a significant diversity between studied individuals. Of the 24% shared haplotypes, less than 25% were shared transregionally, the rest were within regions and more specifically within specific countries. Further, most of the haplotypes were one mutation step away from one another, suggesting recent linkages. 

Our sequences yielded three major haplogroups (A, B and C). (Figure 2, Appendix A). Sub-haplogroup B1 comprised all individuals from West Africa and some from East and Northern Africa, except South African sequences which formed sub-haplogroup B3, suggesting a significant relationship between sheep populations of these regions. Interestingly, sub-haplogroup B1 comprised individuals from all regions except individuals from South Africa, thus suggesting an expanding population and/or admixture. Further, all other haplotypes originating from the central haplotype were one or two mutation steps away. On the other hand, sub-haplogroup B2 consisted of sheep from North (Morocco, Egypt and Algeria) and East African (mainly Kenyan) populations. A mixture of haplotypes in these groups suggests a significant exchange of haplotypes and/or a recent biological group [4]. As expected, the smallest and isolated sub-haplogroup B3 comprised ancient sheep breeds from South Africa. It is noteworthy that haplotype clustering did not follow the geographical distribution.

### 3.3. Neutrality Test

To understand more about the historical background of these populations, we performed Tajima’s D and Fu’s FS values [46] to distinguish between neutrally evolving sequences and sequences evolving under directional selection. Significant negative values suggest a population sub-division or recent population bottleneck, whereas significant positive values indicate a population sub-division [47]. Further, significant negatives imply that the population harbors an excess of rare haplotypes above that expected under neutrality [48]. We observed, as indicated in Table 2, that none of the populations had a significant positive nor negative Tajima’s D, however, the North Africa and West Africa populations had significant negative Fu’s FS tests. Fu’s FS test is considered more sensitive than Tajima’s D. The demographic dynamics of each population and all the sequences pulled together as one population were assessed by use of mismatch distribution analysis, see Appendix A; North and East African sequences displayed multimodal patterns while West African sequences showed unimodal mismatch patterns.

### 3.4. Population Structure

To establish the source of mtDNA CR variation, we performed an analysis of molecular variance (AMOVA) [49,50]. We observed that 54.696% of the variation was accounted for among populations while 45.304% was from within populations an indicator of a strong population structure in African sheep breeds (Table 3). Further, the fixation index from haplotype frequencies was also calculated and the results suggested a lower F_ST_ of 0.016, with a within 98.43% accounting for within-population haplotype frequency (Table 4).

The pairwise F_ST_ between breeds varied significantly, ranging from 0.098 between East Africa and North Africa to 0.986 between South Africa and West Africa (Appendix A). Overall, the fixation index (F_ST_) was high at 0.547.

## 4. Discussion

Autochthonous sheep breeds are vital to the economy and subsistence of many farm families and pastoralists over the African continent. They thrive well in ecologically marginal zones like mountains, arid and semi-arid areas (ASALS) where other domestic animals can not economically thrive [51], and are thought to harbor novel genomic variants that enable them to be resilient to stressful environmental conditions. However, their existence is threatened by the emergence of superior performance breeds. Thus, they are neglected, breeding without directional selection and hence encouraging high levels of inbreeding, which could lead to a loss of genetic diversity. Although their performance is low, these breeds could be crucial in the future, especially when the world is faced with climate change challenges [3].

Understanding the history and phylogenetic relationships of native sheep breeds in Africa is fundamental for the development of breeding and conservation strategies. Several archaeological studies are in concurrence that sheep were among the first animals to be domesticated [16]. Further, population geneticists concur that the present African sheep breeds are mainly descendants of two main ancestors, as inferred by the widespread of two major haplogroups (A and B) [4,29,52]. However, the general genetic history of African continent sheep populations and the relationship among them is not well addressed. The scope of the present meta-analysis was to elucidate their mtDNA CR diversity and phylogenetic relationships as wells as the demographic history of African sheep breeds clustered in four regions.

A high level of mtDNA CR diversity was observed in all studied populations. Results observed in the present study are comparable to findings observed in other research works in different livestock species. For example, Revelo et al. [7] observed a Hd and pi of 0.920 and 0.010 respectively in the Colombian Creole sheep. Likewise, a Hd of 0.998 and pi of 0.250 were observed in Egyptian sheep [20], Moroccan sheep [21], and Arabian goats [9], while Tibetan sheep [29] and Nigerian cattle [53] also exhibited similar results of high diversity among populations. Global organizations also recognize the crucial role of maintaining high biodiversity levels. Genetic diversity as one of the components of biodiversity has been addressed in several international conferences, e.g., [54]. 

Generally, the larger number of haplotypes observed coupled with a higher number of unique haplotypes suggests that a high level of mtDNA CR differences exist among studied individuals and populations. None of the studied regional populations exhibited significantly negative or positive Tajima’s D tests. However, Fu’s FS test returned significant negatives for both the North and West African populations, which could be a suggestive indicator of an expanding population. Mismatch distribution produced a multi-modal pattern for North Africa (Appendix A) and a unimodal for West Africa (Appendix A), agreeing with Fu’s FS test results. However, the East African (Appendix A) population produced an irregular multi-modal pattern contrary to the neutrality test results. In this case, our interpretation suggests that East African sheep could be undergoing selective pressure mainly driven by adaptation to unfavorable production environments. A similar mismatch distribution pattern was also observed for all sequences pulled together (Appendix A). Hence, we can conclude that in general, African sheep populations have undergone selective signatures for adaptation concurring with the study of the world’s sheep breeds which detected genomic regions under selection [55]. It is important to note that processes leading to environmental adaptation could also impact the phenotypic characteristics of an animal and this hypothesis could explain the variation of phenotypes among sheep breeds. The unimodal with a slight skewness pattern of West Africa supported by a significant negative Fu’s FS suggests a recent expansion from a relatively small population, concurring with a narrative that West African populations dispersed further to the Canary Islands and the Caribbean region [56]. The North African sheep population could also have undergone a series of population expansions owing to the archeological history of the role of Egypt in sheep arrival to Africa. For instance, studies indicate that sheep and goats first arrived in Egypt via the Sinai peninsula, the Red Sea and the Mediterranean seacoast, then dispersed Southwards towards Sudan and Ethiopia through the Nile basin [57]. This could explain the haplotypes observed being shared between North and East African sheep populations. Tarekegn et al. [57] also observed shared haplotypes between Ethiopian, Kenyan and Egyptian goats, suggesting a common maternal history.

The median-joining network of haplotypes formed three haplogroups (A, B and C) lacking phylogeographic clustering. Haplogroup B (B1, B3 and B3) was the most common comprising sequences from all studied regions. However, haplogroup A had breeds only from Egypt while haplogroup C consisted of sequences from all North African countries (Morocco, Algeria and Egypt). This observation suggested that studied sheep populations originated from a common matrilineal ancestor (European mouflon) and differentiated into different breeds adapting to local environmental conditions through natural and human selection. Populations in sub-haplogroup B1 are more specifically from West Africa and Sudan (most frequent in hap 233) seem to have displayed further dispersal characteristics compared to other populations in different haplogroups. On the other hand, Sudan played a crucial role in sheep dispersion, as discussed later in this section. Sub-haplogroup B3 formed by the present sequences exclusively included sheep from South Africa. This was expected, as the sequences were obtained from samples excavated from the second layer of the stone age, which is estimated to be more than 2700 years of age [58] and thus could not suggest a conclusive relationship between South African sheep breeds with breeds from other regions. Lack of phylogeographic clustering of small ruminant populations appears to be common; the phenomenon has been observed worldwide in sheep and goats [55,57,59]. 

The African continent is endowed with sheep breeds and varying phenotypes, among which is tail type. Recent research materials have indicated that the fat tail phenotype is an adaptation to hot and dry environments. In our phylogenetic network, sheep breeds characterized with thin tails mainly dominated haplogroup B1 e.g., OulledDjella, Sardi, D’man and all the West African and Sudan sheep breeds. This could be an indication of a common historical background. According to Muigai [16], thin-tailed sheep could have been the first sheep to enter Africa between 7500 and 7000 before present (BP) through the Isthmus of Suez and/or the southern Sinai Peninsula, followed by fat-tailed sheep through the Northeastern part and the Horn of Africa. The fat tail sheep breeds dominated sub-haplogroup B2 (the largest group). The present observations concur with assertions that fat-tailed sheep are widely distributed in Africa, and it is hypothesized that the phenotype developed from selection among thin-tailed sheep breeds [16]. Most of the cross-boundary shared haplotypes, together with the most frequent haplotype (233), were found in sub-haplogroup B1 supporting a narrative of possible Southward dispersion of sheep into Ethiopia and Sudan. The ease with which small ruminants can be transported, their use as items of trade and socio-cultural exchange, and their inherent ability to adapt to a diverse range of production and ecological environments could explain their lack of phylogeographic structure and high level of genetic diversity [57].

The higher variance (54.696%) among populations as compared to within populations (45.304%) suggested the existence of a possible geographical population structure, although this is weak. These results are comparable to other studies that registered similar findings e.g., [7,50,60]. In contrast, other studies observed higher within population variance than among the populations, e.g., [18,47,52]. On the other hand, the difference in haplotype frequencies could be explained by 98.43% within populations. This could be due to the high number of private haplotypes observed and even for shared haplotypes, a larger percentage was within populations. In domestic livestock genetic studies, it is advised to maintain a higher genetic variation within than between breeds [52] because it provides an opportunity for breed improvement to be initiated from within breed selection [60,61]. Higher variation within than among populations suggests a high female-mediated gene flow [52]. The overall F_ST_ value observed was also very high (0.547), indicating high levels of population differentiation. This could be due to the vast geographical representation and vast difference in production environments. Furthermore, the haplotype frequency F_ST_ observed was high (0.016), indicating that haplotype differentiation among populations has occurred but not as much. In general, F_ST_ is regarded as low when the value is below 0.050, moderate when it is between 0.050 and 0.150, high between 0.150 to 0.250, and very high when the value is above 0.250 [60]. Based on the present results, there may be a significant gene reserve in African sheep populations and thus, a need to enhance studies on these populations for a better understanding of their diversity, history, and phylogenetic relationships. 

## 5. Conclusions

The African continent holds an abundant population of autochthonous sheep breeds whose potential has not been fully explored due to inadequate studies conducted on them. mtDNA D-loop marker is one of the very crucial molecular markers that have the potential to provide adequate information on the archaeological history and phylogenetic relationships of indigenous sheep. 

In the present study, the diversity and phylogenetic relationships of 399 sequences of African sheep populations mtDNA CR were meta-analyzed. The observed large number of haplotypes, together with more than 70% unique haplotypes, high nucleotide and haplotype diversity, suggests a very high variability of the mtDNA control region. On the other hand, the median-joining network resulted in the formation of haplotype clusters of mixed populations, suggesting a common matrilineal origin for the sequences studied. Neutrality tests and mismatch distribution analysis indicate a series of population expansions in North and West Africa, and a selection for adaptation in East African sheep populations. Further continental population mismatch analysis pattern suggests selection pressure mostly driven by climate adaptation.

The information provided in this study is vital for the improvement of sheep genetic resource conservation and breeding strategies for the long-term attainment of food security. More studies on archeological history and relationships among African sheep populations diversity and relationships on the recent populations are needed. In addition, the genetic and phenotypic characterization of indigenous sheep populations from different climatic zones as a potential factor for adaptation to local environmental conditions is also recommended.

## Figures and Tables

**Figure 1 biology-10-00762-f001:**
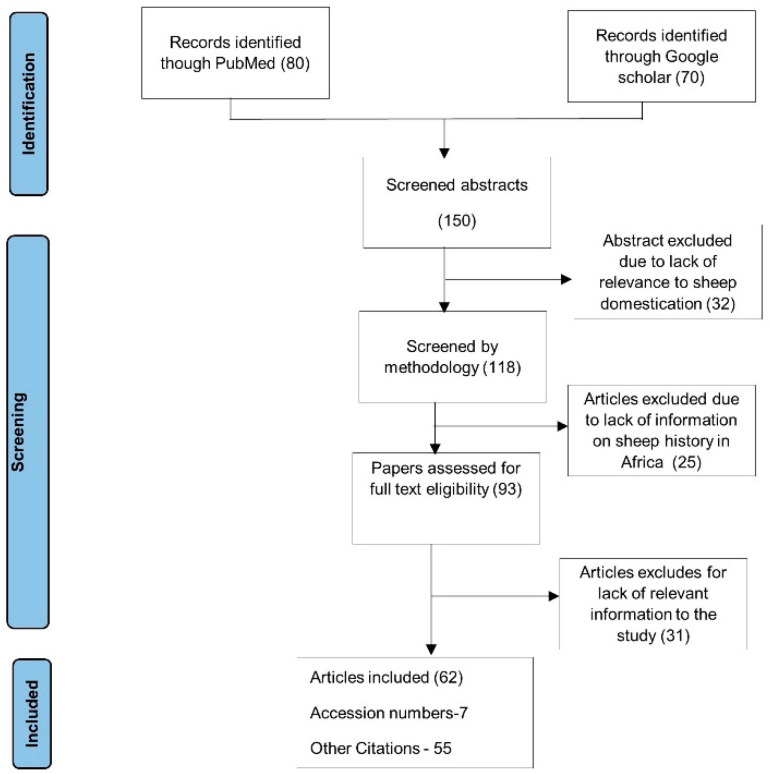
PRISMA (Preferred Reporting Items for Systematic Review and Meta-Analysis) flow chat.

**Figure 2 biology-10-00762-f002:**
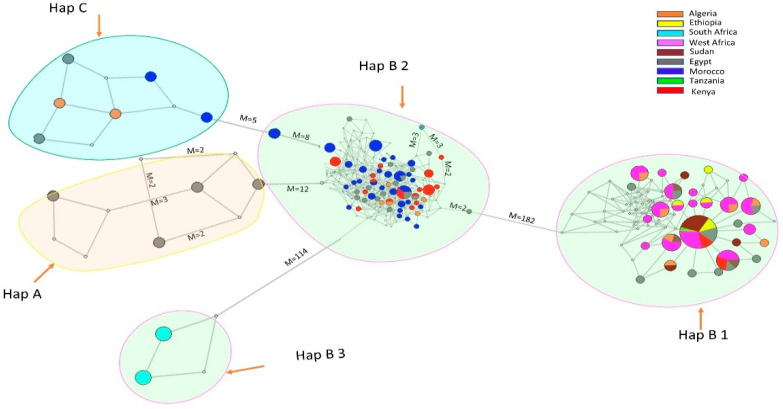
Median-joining network of haplotypes. Circles represent haplotypes, area of the circle is proportional to frequency, colors represent the country, M represents the number of mutations, and links without indicated mutations represent one mutation step.

**Table 1 biology-10-00762-t001:** A summary of polymorphic indices of African sheep breeds.

	N	P	M	Pi	K	H	HD	G + C
1	244	278	351	0.131 ± 0.017	47.436	205	0.998 ± 0.001	0.304
2	44	22	23	0.008 ± 0.001	3.615	29	0.977 ± 0.010	0.418
3	101	290	320	0.263 ± 0.015	116.994	71	0.977 ± 0.008	0.337
4	10	12	12	0.009 ± 0.002	3.578	8	0.956 ± 0.059	0.405
Total	399	284	401	0.254 ± 0.012	84.170	272	0.993 ± 0.002	0.321

1, North Africa (Berbere, Ouled Djellal, Rembi, Rahmani, Ossimi, Barki, Sardi, Romanov, Blanche de Montagne, Boujaad, D’man, Timahdite, Beni-Guil); 2, West Africa (not known); 3, East Africa (Red Maasai, Black Head, East African fat tail, Dorper, Zanziberi, Kabashi, Afar); 4, South Africa (South African Ancient); N, number of sequences; P, number of variable sites; M, total number of mutations; pi, nucleotide diversity per site; k, average number of nucleotide differences; H, number of haplotypes; HD, Haplotype diversity.

**Table 2 biology-10-00762-t002:** A summary of the neutrality test.

Test	Statistics	1	2	3	4	Mean	S.D.
Tajima’s D test	Tajima’s D	0.029	−0.921	3.715	−0.707	0.529	2.163
	*p*-value	0.621	0.197	0.998	0.262	0.515	0.368
Fu’s FS test	FS	−24.003	−10.426	5.659	−1.303	−7.518	12.812
	*p*-value	0.000	0.000	0.938	0.217	0.229	0.465

1, North Africa; 2, West Africa; 3, East Africa; 4, South Africa.

**Table 3 biology-10-00762-t003:** Average F statistics overall loci.

Scheme	D.F.	Sum Squares	Variance Components	Percentage Variation
Among populations	3	9923.905	44.785	54.696
Within populations	395	14,652.210	37.095	45.304
Total	398	24,576.110	81.880	

Fixation index F_ST_: 0.547; *p* value = 0.000.

**Table 4 biology-10-00762-t004:** Fixation index from haplotype frequencies.

Source of Variations	Fixation	Index	F_ST_	Percentage of Variations
Among populations	3	3.195	0.008	1.570
Within populations	395	194.402	0.492	98.430
Total	398	197.596	0.500	

Fixation index F_ST_: 0.016; *p*-value = 0.000.

## Data Availability

Sequences used in this meta-analysis were retrieved from the GenBank database as detailed in Appendix A.

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
