# Peer review of "Meta-Analysis of Mitochondrial DNA Control Region Diversity to Shed Light on Phylogenetic Relationship and Demographic History of African Sheep (Ovis aries) Breeds"

_biology, 2021, doi:10.3390/biology10080762_

Round 1

Reviewer 1 Report

The authors report a meta-analysis of the diversity and phylogenetic relationship of 399 sequences of African sheep populations mtDNA CR. The article is interesting, I think it represents a topic of interest for the scientific community and presents a good data mining methodology, since they used various approximations to answer their question.

I think in this case the paper requires minor revision. Specific comments below:

In the paper you use Mt DNA cr, but it would be better to use it as mtDNA CR since they are more used acronyms and are already standardized

Mismatch distribution and neutrality test indices (Tajima's D values and Fu's Fs) were not included in the methodology, which I imagine were computed using the software Arlequin or DNAsp. Please report the results of mismatch distribution in the results section as well, you can even put the table (Tau (τ), Theta0 (θ0), Theta1 (θ1), SSD and p-value) in supplemental information with the obtained values.

Line 124, remove retrieved from https://www.ncbi.nlm.nih.gov/nucleotide/. This can be put into methodology

Line 137, title and abstract, no capitalization necessary 

Line 138, It is not clear what is referred to as by method, the criteria used may be more specific

Line 179, It would be important to add supplemental information with the list of haplotypes obtained and to which sequence and geographic region they correspond (you can use an acronym) in case the reader wants to contrast the information that they are describing to us.

Line 221, It is not necessary to say the program that is used again, it is already in the methodology, remove

Line 245, change Climate by climate, no capitalization necessary

Line 352, change meta-analysis by study or research, avoid using similar words in the same paragraph

Additionally, the use of the MIGRATE3.6 program (Beerli 2009) is suggested to test different models of historical dispersion, to estimate the levels of gene flow between localities and population size.

Author Response

Response to Reviewer 1

Comments and Suggestions for Authors

The authors report a meta-analysis of the diversity and phylogenetic relationship of 399 sequences of African sheep populations mtDNA CR. The article is interesting, I think it represents a topic of interest for the scientific community and presents a good data mining methodology, since they used various approximations to answer their question.

I think in this case the paper requires minor revision.

Specific comments below:

In the paper you use Mt DNA cr, but it would be better to use it as mtDNA CR since they are more used acronyms and are already standardized

Mismatch distribution and neutrality test indices (Tajima's D values and Fu's Fs) were not included in the methodology, which I imagine were computed using the software Arlequin or DNAsp. Please report the results of mismatch distribution in the results section as well, you can even put the table (Tau (τ), Theta0 (θ0), Theta1 (θ1), SSD and p-value) in supplemental information with the obtained values.

Answer

Thank you so much for the reminder that we had missed describing Mismatch distribution and neutrality test indices (Tajima's D values and Fu's Fs). We have now included them in the methods section as recommended. However, concerning Tau (τ), Theta0 (θ0), Theta1 (θ1), SSD and p-value, since we did not discuss them in our study, we thought it was wise to leave them out. However they are available from corresponding author by request.

Line 124, remove retrieved from https://www.ncbi.nlm.nih.gov/nucleotide/. This can be put into methodology

Answer

We agree with your remark, and as a result we corrected this in the manuscript.

Line 137, title and abstract, no capitalization necessary 

Answer

We agree with your remark, and as a result we corrected this in the manuscript.

Line 138, It is not clear what is referred to as by method, the criteria used may be more specific

Answer

We agree with your remark, and as a result we corrected this in the manuscript to mean screened by markers because our main interest was partial mtDNA control region sequences.

Line 179, It would be important to add supplemental information with the list of haplotypes obtained and to which sequence and geographic region they correspond (you can use an acronym) in case the reader wants to contrast the information that they are describing to us.

Answer

We agree with your remark, and as a result we included an excel supplementary Table S3.

Line 221, It is not necessary to say the program that is used again, it is already in the methodology, remove

Answer

Thank you for your remark, and as a result we corrected this in the manuscript.

Line 245, change Climate by climate, no capitalization necessary

Answer

It was corrected in the manuscript.

Line 352, change meta-analysis by study or research, avoid using similar words in the same paragraph

Answer

We agree with your remark, and as a result we corrected this in the manuscript.

Additionally, the use of the MIGRATE3.6 program (Beerli 2009) is suggested to test different models of historical dispersion, to estimate the levels of gene flow between localities and population size.

Answer

We take this opportunity to thank you for suggesting MIGRATE 3.6 program. By quick perusal it seems a useful software in determining historical gene between localities as indicated in your comments above. Our main scope was to examine the relationship between the populations and use the existing literatures to explain possible causes as per the reviewed literatures. We hope to use this software in our future analyses. Thank you again for the suggestion.

Reviewer 2 Report

The authors of study entitled “Meta-analysis of Mitochondrial DNA control region diversity phylogenetic relationship and demographic history of African sheep (Ovis aries) breeds” aimed to evaluate the genetic diversity and demographic history of 399 sequences of African indigenous sheep breeds from previously published research articles. The paper is not so fluently written, except for the Discussion section, which testifies that authors accurately read up on published literature.

Major comments:

The title of the manuscript is not sliding. I think you should modify it, for example: “Meta-analysis of Mitochondrial DNA control region diversity to shed light on phylogenetic relationships and demographic history of African sheep (Ovis aries) breeds”.

The evident lack of commas throughout the text makes the reading not fluent. Please, add commas whenever it is needed.

The authors used “Mt DNA” to indicate the “mitochondrial DNA”, but you should correct it throughout the text with “mtDNA”. Please modify.

At the end of the Introduction, the authors state that “Similar but few studies have been conducted in Africa with the main aims of as sessing the evidence of domestication and inferring the ancestral origin of the sheep breeds by observing the haplogroups present in the populations. However, to the best of our knowledge, there is no literature exploring the phylogenetic relationship among the sheep breeds in the African continent.”. I think that the manuscript completely lacks of a discussion concerning the haplogroup classification, a crucial point when analysing mtDNA data. I think it is one of the weak links of this work and the authors should improve the paper by adding a table with haplogroup and cluster information and geographic area (also country, not only the four regions you have arbitrarily decided to separate the African continent) for each sequence here analysed, and discuss the haplogroup variation (and frequencies) among African sheep breeds.

I also think that in the table you should indicate the name of each breed/population included in the study.

Figure 1: please, better specify why you excluded some articles.

Figure 2: is there a correlation between your clusters and the mtDNA haplogroups from literature? I think you should indicate also the haplogroups in the network.

Minor comments:

Line 11: “Continent” with capital letter?

Line 16: replace “Understanding the” with “Knowledge about”.

Lines 17: modify with “relationships” and replace “among” with “between”.

Lines 19: replace “MtDNA” with “mitochondrial DNA”.

Line 19-20: delete “mitochondrial D-loop”.

Line 21: delete “sheep” before “genetic resources”.

Line 24: replace “observed” with “found”.

Line 25: “were unique and a high….”. Delete “also”.

Line 26-27: modify “and haplotype diversity observed was high; 284, 0.254 ± 26 0.012 and 0.993 ± 0.002, respectively.” with “and haplotype diversity were high (284, 0.254 ± 26 0.012 and 0.993 ± 0.002, respectively)”.

Line 28: what do you mean with “universal”? Please explain.

Line 29: delete “or”.

Line 32: “larger sample” or “more samples”.

Line 33: replace “higher” with “deeper”.

Line 40: “development goal 2”?.

Line 41: “goat and cattle populations…”. Move the reference “[2]” after “respectively”.

Line 43-44: move “researchers” after “decades”.

Line 44: delete “the”.

Line 49: “…suggests that sheeps are…”.

Line 53: “… the raise of mitochondrial haplogroups…”. Move “globally” before “most”.

Line 57: “Sardinia”.

Line 59: “…sheep clades from C to E.”.

Line 67: “lineages”.

Line 70: delete “also”.

Line 74: “clades”.

Line 76: replace “studied in” with “from”.

Line 78: replace “have” with “belong to”.

Line 82-83: replace “other haplogroups which is also an area to be explored in future studies” with “other haplogroups, an area to be explored in future studies”.

Line 91: “polymorphisms”.

Line 92: delete “and” after “relationships”.

Line 93: replace “use of” with “by using”.

Line 99: add “been” before “proved”.

Line 100-102: please re-write the sentence.

Line 105: delete “on” before “the origin”.

Line 108: “This could be the evidence of a post domestication selection…”.

Line 120: “relationships”.

Line 123-124: replace “….analyzing the Mt DNA control region sequences from the NCBI GenBank database [36] retrieved from https://www.ncbi.nlm.nih.gov/nucleotide/.” with “…analyzing the mtDNA control region of 399 sequences retrieved from the NCBI GenBank database [36].”.

Line 137: modify with “The articles were screened by title and….”.

Line 139-142: did the authors consider only modern mtDNA sequences or also ancient? Please specify in the manuscript.

Line 145: “flow chart”.”

Line 147: “481 base pairs”, please indicate the range (from nucleotide position … to np …) you analysed.

Line 150-151: please specify which countries belong to North Africa, which from West Africa, etc.

Line 166: add “respectively” after “0.002”.

Line 175: replace with “A summary of polymorphic indices of African sheep breeds”.

Line 181: “…between the studied individuals.”.

Line 186: “Figure 2” between parenthesis.

Line 193: move “On the other hand” at the beginning of the sentence.

Line 193: please indicate how many haplotypes belong to this cluster, so everyone can understand what you mean with “largest”.

Line 194-196: move the sentence to the Discussion section.

Line 200: “area of the circle…”.

Line 201: “Country” and delete “n”.

Line 214-215: delete “see” and put “Figure S1” between parenthesis.

Line 222-223: “…populations, an indicator of a strong population structure in African sheep breeds (Table 3).”.

Line 225, line 227 and line 229: replace “FST” with “FST”.

Line 226 and line 228: “Table 4” and “Table A2” between parenthesis

Line 246: “relationships”.

Line 247: “…Africa is fundamental for the development…”.

Line 252: “relationships”.

Line 253: “…was to elucidate…”.

Line 256: move “studied” before “populations”.

Line 272: delete “also”.

Line 275: replace “…environments, similar mismatch…” with “environments. A similar mismatch…”.

Line 297: replace “cluster I” with “cluster 1”.

Line 299: “On the other hand, Sudan played a crucial role in sheep…”.

Line 303: “…thus it could…”.

Line 304: replace “with” with “and”.

Line 325: “The higher variance…”.

Line 326-327: replace “(45.304%) was observed suggesting the existence of population structure possibly geo-326 graphical although week.” with “(45.304%) suggested the existence of a possible geographic population structure although week.”.

Line 335: “…suggests a high female…”.

Line 336: put “0.547” between parenthesis.

Line 347: delete “the” at the beginning of the sentence.

Line 352: “relationships”.

Line 354: “large number” not “larger”.

Line 355-356: replace “…suggests high variability of Mt DNA control region. The Median-joining network on the other hand resulted in the formation…” with “suggests a very high variability of mtDNA control region. On the other hand, the Median-joining network resulted in the formation”.

Line 357: “…origin for the studied…”.

Line 360: “…suggests a selection…”.

Line 366: “…indigenous sheeps from…”.

Author Response

Response to Reviewer 2

Comments and Suggestions for Authors

The authors of study entitled “Meta-analysis of Mitochondrial DNA control region diversity phylogenetic relationship and demographic history of African sheep (Ovis aries) breeds” aimed to evaluate the genetic diversity and demographic history of 399 sequences of African indigenous sheep breeds from previously published research articles. The paper is not so fluently written, except for the Discussion section, which testifies that authors accurately read up on published literature.

Major comments:

The title of the manuscript is not sliding. I think you should modify it, for example: “Meta-analysis of Mitochondrial DNA control region diversity to shed light on phylogenetic relationships and demographic history of African sheep (Ovis aries) breeds”.

Answer

We fully agree with esteemed Reviewer’s remark, and as a result we corrected this in the manuscript.

The evident lack of commas throughout the text makes the reading not fluent. Please, add commas whenever it is needed.

Answer

We have carefully revised the manuscript and together with the suggestions given by both reviewers, we hope that the revised version meets the English standards expected.

The authors used “Mt DNA” to indicate the “mitochondrial DNA”, but you should correct it throughout the text with “mtDNA”. Please modify.

Answer

We agree with your remark, and as a result we corrected this in the manuscript.

At the end of the Introduction, the authors state that “Similar but few studies have been conducted in Africa with the main aims of as sessing the evidence of domestication and inferring the ancestral origin of the sheep breeds by observing the haplogroups present in the populations. However, to the best of our knowledge, there is no literature exploring the phylogenetic relationship among the sheep breeds in the African continent”. I think that the manuscript completely lacks a discussion concerning the haplogroup classification, a crucial point when analysing mtDNA data. I think it is one of the weak links of this work and the authors should improve the paper by adding a table with haplogroup and cluster information and geographic area (also country, not only the four regions you have arbitrarily decided to separate the African continent) for each sequence here analysed and discuss the haplogroup variation (and frequencies) among African sheep breeds.

Answer

We agree with your remark, and as a result we corrected this in the manuscript.

I also think that in the table you should indicate the name of each breed/population included in the study.

Answer

Although it is not clear which table you were referring, but we hope all missed information are available from Table S3. It was our wish to include individual populations, but West African sequences could not be segregated into individual populations as in the database, they are referred to as West Africa. Therefore, for uniformity purposes, all other sequences were referred to as per the region.

Figure 1: please, better specify why you excluded some articles.

Answer

We agree with your remark, and as a result we corrected this in the manuscript (Figure 1).

Figure 2: is there a correlation between your clusters and the mtDNA haplogroups from literature? I think you should indicate also the haplogroups in the network.

Answer

Since we did not group the sequences in haplogroups, we could not establish the haplogroup correlation, however, multiple clustering showed a certain level correlation.

Minor comments:

Thank you for the esteemed Reviewer’s comments. All remarks, suggestions were accepted and corrected in the manuscript. It was carefully revised for syntax-errors, misspellings and not appropriate words by the authors once again.

Round 2

Reviewer 2 Report

The authors modified the manuscript as suggested and it is now suitable for publication. I have only very few writing mistakes to report.

Please, correct as follows:

L148: "sequences"

L195: “haplogroups (A, B and C)"

L197: "Sub-haplogroup B1"

L199: “sub-haplogroup B3""

L200: "sub-haplogroup B1 comprised"

L201: "South Africa, thus suggesting"

L206: "sub-haplogroup B2"

L209: "sub-haplogroup B3"

L210: "comprised ancient sheep breeds from South Africa"

L310-311: " consisted of sequences from all North African countries"

L315: “sub-haplogroup B1"

L317: “other populations in different haplogroups"

L319: "Sub-haplogroup B3"

L329: “sub-haplogroup B1"

L335: “sub-haplogroup B2"

Author Response

We are grateful for the esteemed Reviewer' efforts done during the manuscript's revision. It was carefully revised for syntax-errors, misspellings and not appropriate words by the authors once again.